# Boron Nitride Nanoribbons Grown by Chemical Vapor Deposition for VUV Applications

**DOI:** 10.3390/mi13091372

**Published:** 2022-08-23

**Authors:** Jiandong Hao, Ling Li, Peng Gao, Xiangqian Jiang, Chuncheng Ban, Ningqiang Shi

**Affiliations:** 1MEMS Center, Harbin Institute of Technology, Harbin 150001, China; 2Key Laboratory of Micro-Systems and Micro-Structures Manufacturing, Ministry of Education, Harbin 150001, China; 3Solar Cell Research Laboratory, Tianjin Institute of Power Sources, Tianjin 300381, China

**Keywords:** VUV detectors, wide-bandgap semiconductors, boron nitride nanoribbons, chemical phase deposition

## Abstract

The fabrication process of vacuum ultraviolet (VUV) detectors based on traditional semiconductor materials is complex and costly. The new generation of wide-bandgap semiconductor materials greatly reduce the fabrication cost of the entire VUV detector. We use the chemical vapor deposition (CVD) method to grow boron nitride nanoribbons (BNNRs) for VUV detectors. Morphological and compositional characterization of the BNNRs was tested. VUV detector based on BNNRs exhibits strong response to VUV light with wavelengths as short as 185 nm. The photo–dark current ratio (PDCR) of this detector is 272.43, the responsivity is 0.47 nA/W, and the rise time and fall time are 0.3 s and 0.6 s. The response speed is faster than the same type of BN-based VUV detectors. This paper offers more opportunities for high-performance and low-cost VUV detectors made of wide-bandgap semiconductor materials in the future.

## 1. Introduction

In addition to radiation from the sun, humans can produce Ultraviolet (UV) light for medical treatment, disinfection of “sun kitchens”, UV curing, UV communication, and missile tail flames, etc. [1,2]. Vacuum Ultraviolet (VUV, 10–200 nm) detectors can convert VUV radiation signals into electronic signals, so that VUV radiation can be observed easily, detected accurately, and effective warnings can be made early. They have a wide range of applications in both civilian and military fields. For example, solar storm observation, deep space communication and planetary atmosphere re-cording, etc. [3,4,5,6].

With the development of science and technology, humans have certain requirements for VUV detectors, such as smaller size, faster response speed, lower fabrication cost, more stable performance, and green environmental protection, etc. A new generation of VUV detectors must be developed to meet these growing demands. Modern VUV detectors have the characteristics of high stability, high sensitivity, fast speed, and high signal-to-noise ratio. In recent years, VUV detectors based on a wide-bandgap semiconductor has become a research focus with the progress of semiconductor material preparation technology. The fabrication process of VUV detectors based on silicon materials and other conventional III-V compound semiconductor materials has been relatively mature. These materials have narrow bandgap, and detectors operating in the VUV region must be equipped with filters, which will increase the overall cost of the detector. Therefore, the choice of detectors’ materials has turned to the new generation of wide-bandgap semiconductor materials. As a third-generation semiconductor material, boron nitride (BN) is a synthetic ultra-wide bandgap semiconductor material with a forbidden bandgap greater than 6.0 eV, which is a relatively wide bandgap in wide-bandgap semiconductor materials. Its intrinsic absorption edge is about 207.5 nm, and its absorption coefficient is near the absorption edge, as high as 7 × 10^5^ cm^−1^. Its dielectric strength is as high as 8 MV/cm. BN nanomaterials exhibit outstanding chemical and thermal stability at high temperatures, and have become one of the most widely studied nanomaterials. Therefore, the fabricated device can maintain stable operation under poor conditions. It means that BN is suitable for the fabrication of VUV detectors [7,8,9,10,11,12,13,14,15]. BN nanomaterials can be divided into boron nitride nanoribbons (BNNRs), boron nitride nanotubes (BNNTs), etc. BNNTs are hollow tubular structures, and BNNRs are curved ribbon-like structures. At present, BNNTs are widely used. However, BNNRs are rarely used due to their complex preparation process and low yield. BNNRs also have a larger area that can perceive VUV light, which is also more beneficial to the characteristics of the device [16,17,18].

In this work, we use the chemical vapor deposition (CVD) method to grow BNNRs in situ. The characterization results show that the BN grown by this method has high crystalline quality. The VUV detector based on BNNRs has good detection performance under 185 nm light with a photo–dark current ratio (the maximum ratio of photocurrent to dark current, PDCR) of 272.43, a responsivity (R) of 0.47 nA/W, rise time (T_r_) of 0.3 s, and fall time (T_f_) of 0.6 s. The speed is faster than the same type of BN-based VUV detector. The low-cost VUV detector based on BNNRs which fabricated in this work has great potential for application.

## 2. Experimental Section

### 2.1. Preparation of Boron Nitride Nanoribbons

The preparation process of BNNRs has been described in a previous study [19]. BNNRs were grown in situ by CVD method, that is, mixing alkali metals with boron nitride nanotubes (BNNTs). The alkali metal particles embedded in BNNTs were mixed with NH_3_ at high temperature. During the reaction, the resulting material directly causing the relatively weak covalent bonds to expand and exert tension on the surrounding atoms, eventually decompressing BNNTs to form BNNRs [20,21]. The B powder (Aladdin, purity 95%) was mixed with Li_2_O powder (Aladdin, purity 97%) under N_2_, and the molar ratio of Li_2_O to B was controlled at 0.15:1. The mixed powder is placed in the alumina quartz boat and sintered for 1.5 h at 1150 °C in a single-tube high-temperature furnace (Model: 3216). The powder is removed after cooling to room temperature. NH_3_ is introduced at a rate of 0.2 L/min for reaction during the whole process. BNNRs were stripped by mechanical stripping. The sample is shown in Figure 1a.

### 2.2. Device Fabrication

We use a glass substrate for good light transmission. Firstly, Au electrodes were sputtered on clean glass substrates by magnetron sputtering through an interdigital electrode mask [22,23,24]. Then, we dispersed BNNRs powder in alcohol solution at a volume ratio of 0.5:1, took it out and let it stand for 5 min after shaking with an ultrasonic oscillator for 0.5 h. The supernatant of the solution was drop-coated on Au interdigital electrodes and dried in an oven for 30 min. The fabricated VUV detector is shown in Figure 1d.

## 3. Results and Discussion

### 3.1. Characterization of Boron Nitride Nanoribbons

The prepared powder was characterized by SEM which had many needle-like structures, as shown in Figure 1b. BNNRs and BNNTs can be observed in Figure 1c after magnification. BNNTs are marked as square and BNNRs are marked as oval. It can be verified that BNNRs were in situ grown by cracking BNNTs with alkaline metal. We can see from Figure 1c that there are many irregular granular materials around BNNRs, which are impurities from the preparation process [25,26].

We also characterized BNNRs by TEM. Figure 1e,f show the TEM and HRTEM images of BNNRs. The ribbon-like morphology of BNNRs can be clearly seen from Figure 1e and the straight and parallel lattice fringes can be seen in Figure 1f, where the inset (S1) is an enlarged lattice fringe. It can be concluded that the lattice spacing is 0.25 nm. The inset (S2) is the electron diffraction pattern, and it can be inferred that the BNNRs are h-BN with good crystallization [27].

To further study the powder, we used X-ray photoelectron spectroscopy (XPS) to characterize BN, as shown in Figure 2a,b. The obtained N 1s-core level is shown in Figure 2a. The peak at the binding energy is 397.9 eV, which is attributed to the N-B bond structure. The obtained B 1s-core level is shown in Figure 2b. The center of the binding energy peak of B1s is 190.4eV which is caused by the B−N bond, while the peak at a lower binding energy (191.8 eV) is attributed to the B–O bond. This also proves that the reaction product has a small amount of oxide impurities [28,29].

Figure 2c shows the Fourier Transform Infrared Spectroscopy (FTIR) of BN powder. Its vibration modes include transverse optical (TO) mode, longitudinal optical (LO) mode and out-of-plane buckling (R) mode, etc. The absorption peaks of BN are located at 810 cm^−1^ and 1380 cm^−1^, respectively. The two in-plane optical phonon modes of the transverse optical (TO) mode and the longitudinal optical (LO) mode are equal in the X and Y axes at about 1380 cm^−1^, corresponding to the stretching vibration of the B-N atom in the plane. The out-of-plane buckling (R) mode resonates around 810 cm^−1^, corresponding to the out-of-plane bending vibrations of B-N-B atoms [30,31]. There are many irregular particles around BNNRs and BNNTs. Therefore, a small number of B-N-B vibrational modes may arise from the interaction of adjacent particles. FTIR results are consistent with SEM-characterized surfaces [32,33].

Raman spectroscopy was used to characterize the BN powder as shown in Figure 2d. A sharp peak corresponding to the B-N $E_{2g}^{1}$ mode of h-BN was obtained near 1366 cm^−1^. The FWHM of this study is 14 cm^−1^ wider than previous studies. Since the oxides and catalyst derivatives of B element exist in the prepared powder, it is consistent with the above characterization results [34,35]. The results show that the BNNRs prepared in this study have good ribbon morphology, good crystallization and high composition purity, making them are suitable for the application of VUV detectors.

### 3.2. Characterization of the Vacuum Ultraviolet Detector-Based Boron Nitride Nanoribbons

We use the semiconductor analysis and test system (model: B1500) to test the characteristics of the VUV detector based on BNNRs. The wavelengths of VUV light sources used are 185 nm, 254 nm, and 365 nm. Figure 3a shows the *I*−*V* characteristic test curve of the VUV detector based on BNNRs. The photocurrent of the VUV detector is basically the same as the dark current under the light sources with wavelengths of 254 nm and 365 nm. The reason for this is that BN is a wide bandgap semiconductor material with 6.0 eV. Its corresponding intrinsic absorption limit is about 207.5 nm [36]. Therefore, it has weak absorption for wavelengths above 207.5 nm. The photocurrent changes greatly under 185 nm light source, the dark current (*I_d_*) and photocurrent (*I_p_*) are symmetric distribution at the center of origin, and the slope of *I_p_* is larger than that of *I_d_*. This indicates that the electrons in the BNNRs are moving and the resistance of the BNNRs decreases when the light is turned on [37]. Among them, we use a light power meter (VLP-2000) to detect the light power density, which is 1.28 mw/cm^2^. The maximum value of the ratio of photocurrent to dark current is calculated to be 272.43 under 0 V bias. Here, the *PDCR* is:(1)$$PDCR=I_{p}/I_{d}$$
where *I_p_* is the photocurrent, and *I_d_* is the dark current. The maximum value of the ratio of photocurrent to dark current is calculated to be 272.43 under 0 V bias. After that, we changed the light power density of the VUV light to test the *I*−*V* characteristics (0 V bias). The light power density is obtained from the light power meter (VLP-2000). As shown in Figure 3b, the light power density was increased from 0.2 mW/cm^2^ to 1.28 mW/cm^2^, and the inset shows a partially enlarged curve. As the light power density increases, the photocurrent also increases gradually. According to the results of the *I-V* characteristics, the VUV detector based on BNNRs can detect VUV light.

In order to further study the characteristics of the VUV detector based on BNNRs. We use the semiconductor analysis and test system (model: B1500) to measure the *I*−t characteristics with different bias voltages at the same light power density. There are only detectors and semiconductor analysis testers in the test circuit that do not have any resistance. The light source (185 nm, 0.6 mW/cm^2^) is equipped with a shutter that switches on and off with a period of 5 s, as shown in Figure 3c. When V_bias_ = 0 V, the *I*_pmax_ is maintained at a maximum of ~4 pA, and when V_bias_ = 5 V, the *I*_pmax_ is basically maintained at ~6 pA, when *V*_bias_ = 10 V, the *I*_pmax_ remains at ~8 pA. This is consistent with the results of the previous *I*−*V* characteristics tests. The *I*−*V* characteristic curve of the VUV detector is linear, which proves that the resistance value is certain. When the *V*_bias_ increases, the *I_p_* also increases. Figure 3d,e shows the rise/fall time of the VUV detector based on BNNRs, and the current is very sensitive to VUV light. When the VUV lamp is turned on, the current rises rapidly to a high level and then gradually saturates. When the lamp is turned off, the current drops sharply to a low value again. The rise time is 0.3 s and the fall time is 0.6 s. This work is faster than the same type of VUV detector [38,39,40]. Table 1 is a comparison of parameters with previous studies.

The responsivity (*R*) can be obtained from the formula:(2)$$R=\frac{I_{p}}{A\cdot P_{inc}}$$
in which *I_p_* is the photocurrent, *A* is the active area of the device (0.9 cm^2^) and *P_inc_* is the light power density (1.28 mW/cm^2^). The active area of the device refers to the photosensitive area of the VUV detector, that is, the area of the BN film [41]. The active area of the device is calculated according to the scale of SEM which is about 0.9 cm^2^. Light power density is measured by light power meter (VLP-2000). Figure 3f shows the relationship between R and light intensity. The maximum responsivity is 0.47 nA/W under 185 nm. It is not difficult to find that the responsivity increases rapidly when the light density increases. The 6.0 eV wide bandgap of BNNRs corresponds to the energy of incident light with a wavelength of 185 nm. The energy of the incident light excites electrons in the bandgap of BNNRs, and the energy of electrons in the valence band is transferred to the conduction band. The photocurrent recombination loss of the device is reduced at high optical density, which leads to the higher responsivity as the light intensity continues to increase. The good VUV selectivity can ensure that BNNRs-based VUV detectors can detect VUV light without filter structures [42,43,44]. However, the responsivity of the VUV detector based on BNNRs is very low, which is due to fabrication process. We will aim to improve it in the future.

## 4. Conclusions

In conclusion, BNNRs were prepared by CVD method. At the same time, SEM, TEM, XPS, and FTIR characterization tests were used to analyze their morphology and composition. The results show that BNNRs have good ribbon-like morphology and good crystallinity, which is suitable for VUV detectors. The test results of the VUV detector based on BNNRs show that it is sensitive to wavelengths of 185 nm. The VUV detector based on BNNRs has good detection performance with a PDCR of 272.43, R of 0.47 nA/W, T_r_ of 0.3 s, and T_f_ of 0.6 s, which is faster than the same type of BN-based VUV detector. The BNNRs-based VUV detector fabricated in this work is simple in structure and low in cost, which provides a reference for the development of high-performance wide-bandgap semiconductor material VUV detectors. The rise/fall time of the BNNRs-VUV detector is faster than that of the same type. However, the responsivity needs to be improved.

## Figures and Tables

**Figure 1 micromachines-13-01372-f001:**
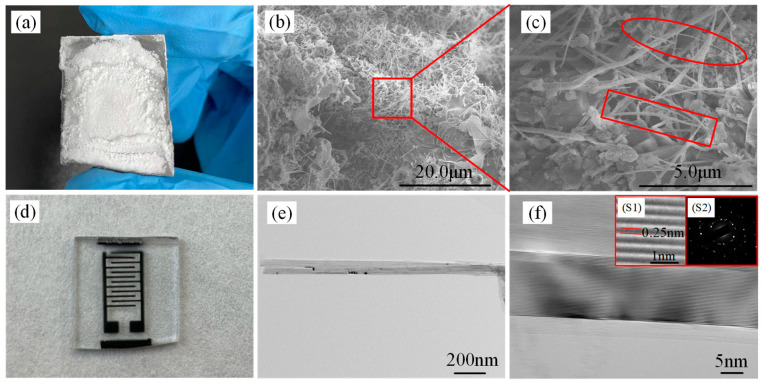
(**a**) BNNRs prepared by CVD method. (**b**) SEM image of BNNRs. (**c**) Typical magnified SEM images of the selected surface area. (**d**) The VUV detector based on BNNRs. (**e**) TEM image of BNNRs. (**f**) Lattice fringes (S1: Lattice fringes magnified locally. S2: Diffraction pattern).

**Figure 2 micromachines-13-01372-f002:**
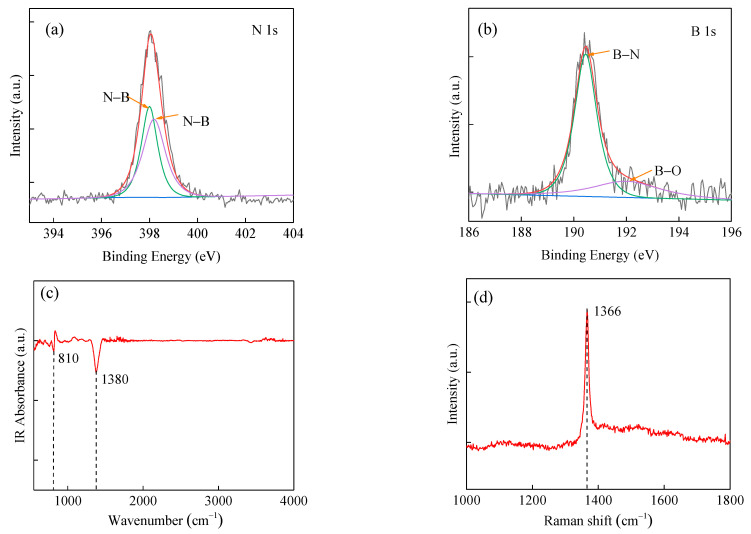
(**a**,**b**) XPS of BNNRs. (**c**) FTIR. (**d**) Raman scattering spectrum.

**Figure 3 micromachines-13-01372-f003:**
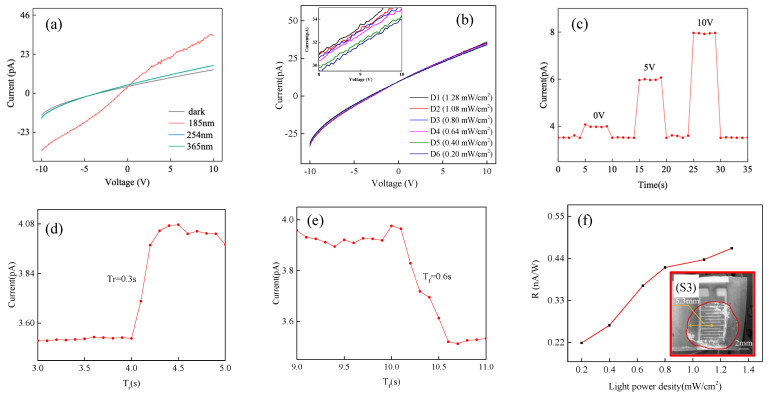
(**a**) *I*−*V* characteristic of the BNNRs-VUV dectector. (**b**) *I*−*V* characteristic of VUV detector with different light power densities. (**c**) *I*−t characteristic under different bias voltages. (**d**) Rise time of the detector. (**e**) Fall time of the detector. (**f**) Responsivity (S3: SEM image of the device. The active area is about 0.9 cm^2^).

**Table 1 micromachines-13-01372-t001:** Parameter comparison of VUV detectors.

Material	Responsivity (mA/W)	Tr/Tf (s)	Refs.
AIN	4.5	8/3	[38]
Diamond films	—	4/0.2	[39]
BN	4.8 × 10^−3^	5.43/1.83	[40]
BN	0.47 × 10^−3^	0.3/0.6	This work

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
