# Peer review of "Boron Nitride Nanoribbons Grown by Chemical Vapor Deposition for VUV Applications"

_micromachines, 2022, doi:10.3390/mi13091372_

Round 1
Reviewer 1 Report
In this study Jiandong Haoand et al proposed a novel DUV detector based on boron nitride nanoribbons. The paper describes the technology of growth of samples by the CVD method. A detailed characterization was carried out by various methods of microscopy and Raman spectroscopy. The responsivity of the detectors and their other characteristics (PDCR, response time) were measured carefully and reported. The low cost makes these detectors very attractive to manufacture.
But there is an important question. Modern DUV-detectors have a responsivity of about 10^7 A/W [http://dx.doi.org/10.1039/C5TC01178G], this is a State-of-the-art [http://dx.doi.org/10.3390/s18072072]. How could the authors improve the resulting responsivity? Please write it in the article.
The work contains a number of typos
- page 3, “Ningqiang shi” instead “Ningqiang Shi”
- page 4, Fig 2(d), “Roman Shift” instead “Raman Shift”
Thus, taking into account minor remarks, I can recommend publication in Micromachines-MDPI.
Reviewer 2 Report
1. The performance of the proposed device does not describe superior detection characteristics compared to the previously reported device, so it is necessary to compare it with the previously reported Deep UV detector for the figures-of-merit (FOM)
2. Since the device geometry used for measurement does not take into accounts the wavelength, intensity of incident light, response speed, and fields of application, there must be a measurement results with the change in device geometry or dimension.
3. The bias condition of PDCR extracted from Fig.3 (a), (c) is required, and the time-scale of the measurement on the device is inappropriate to mention the response time. Measurement data at time-scale of less than msec. should be presented.
4. Since the papers presented as references for device fabrication in the manuscript have low relevance to the proposed BNNR based UV detector in terms of the detectable wavelength range, device structure, and substrate, it is appropriate to substitute the references for the BNNR-based photodetector.
Reviewer 3 Report
This work reports characterization of BNNR based DUV detectors, which could have practical value. However, there are several critical issues with the conclusion that can be drawn from the experiment data. The authors are suggested to either clarify these issues or provide more data to address the concern. Besides, some experiment conditions are not clear, which prevent precise understanding of the results. As responsivity is a critical figure of merit, table 1 shows that the BN in this work has a significantly smaller than other works, which seems to be an unfavorable factor of the merit of this work. lastly, the authors are recommended to make conclusions based on quantitative data, instead of qualitative descriptions. A major revision is necessary before further consideration of publication of this work.
1) Clarification needed. Introduction section. “Its intensity is up to 8mV/cm”. What intensity this is?
Besides, the motivation of studying BNNR should be provided. Why BNNR? Any advantage of BNNR compared to other ultra-wide bandgap semiconductors? What’s the goal of this study?
2) Regarding BNNR vs BNNT. Some background info about their comparison of properties should be provided, which can benefit readers not familiar with these materials. For example, can BNNT be used for DUV detector? How does the ribbon morphology can benefit the detector performance, probably, theoretically?
3) “While the peak at a lower binding energy (191.8 eV) is attributed to the B–O bond”. comments needed for the B-O bond. Does that mean there are oxide impurities in material?
4) “The photocurrent of the DUV detector is basically same as the dark current under the light sources with wavelengths of 254nm and 365nm.” The detector response to different wavelength of UV lights is a critical part of its performance. Analysis on why the detector showed weak response to 254 and 365nm should be provided. Is it due to BN has a bandgap 6eV so that it has a weak absorption to wavelength longer than 207nm?
5) Accurate definition of PCDR is necessary. Apparently, the stronger the light intensity, the larger the PCDR. So at what light intensity the PCDR is defined? Besides, the bias voltage also affect the photocurrent and dark current, so at what bias voltage the PCDR is defined? Without an clear explanation of PCDR, it is not convincing to draw the conclusion “The low Id and high PDCR show that the DUV detector based on BNNRs can be applied to detect weak DUV light.”
6) I did not find an analysis of Fig. 3b in the main text.
7) The experimental setup for the measurement of rise time and fall time should be provided. What instruments are used? How fast is your lamp turning on or off?
8) Fig. 3(d) shows the relationship between R and light intensity. Does Fig. 3(d) needs to be corrected as Fig. 3(e)? Besides, when light power density is calculated, what area is used as the device active area? what area should be used for such calculation according to literatures?
9) In Figure 3f, is this dark IV or IV under light? When the film area becomes larger, both the dark current and photocurrent scale up. So the data in Figure 3f cannot support “the detection ability of the DUV detector is stronger and the sensitivity is higher, when BN film becomes larger”
Round 2
Reviewer 1 Report
I recommend this paper for publication
Author Response
Thanks for your advises.
Reviewer 3 Report
While some of the minor concerns were addressed by authors, unfortunately, the major concerns related to some important conclusions were not properly addressed with solid evidence. Besides, the validity and methodology of some experiments are in doubt.
1) regarding the PDCR. If PCDR is used to support the detector performance is good, then a formal definition is necessary. The authors failed to present a formal definition of the PDCR. Only a simple description of how they calculate the PCDR is given.
“The low Id and high PDCR show that the DUV detector based on BNNRs can be applied to detect weak DUV light.”
If a even stronger light source is used, a higher PCDR will be obtained. Then How high is high? At least this value should be compared with literature values.
Besides, this description is only qualitative. Typically, quantitative figure of merits like Noise Equivalent Power, responsivity, and specific detectivity are used to quantify the detector performance, including detecting faint light. None of these parameters were found in this work.
2) regarding the rise time and fall time. The authors did not respond to “How fast is your lamp turning on or off?”. Although authors mentioned shutter is used, how fast is your shutter turned on and off is not mentioned? The intrinsic time of shutter turning on and off affects the measured detector response time. Besides, the RC constant of the measurement circuit also affects the rise/fall time. These info is also omitted in the manuscript.
3) what area is used as the device active area? what area should be used for such calculation according to literatures?
Authors did not respond to what is the device active area. the area of the BN film is not equal to device area. what area should be used is also not clear
4) In Figure 3f, is this dark IV or IV under light?
Authors did not answer this question. Scientifically, could this difference be due to individual device quality, not area? Some quantitative parameters should be used. When device area become large, the photocurrent density presumably should stay constant, given the same power density, the responsivity should not change.
Round 3
Reviewer 3 Report
Authors have made their efforts addressing the concerns, and it may be published.